# Prediction of In-Hospital Falls Using NRS, PACD Score and FallRS: A Retrospective Cohort Study

**DOI:** 10.3390/geriatrics8030060

**Published:** 2023-06-01

**Authors:** Jennifer Siegwart, Umberto Spennato, Nathalie Lerjen, Beat Mueller, Philipp Schuetz, Daniel Koch, Tristan Struja

**Affiliations:** 1Medical University Clinic, Kantonsspital Aarau, 5001 Aarau, Switzerlandschuetzph@gmail.com (P.S.);; 2Medical Faculty, University of Basel, 4001 Basel, Switzerland

**Keywords:** falls, in-hospital, prediction, post-acute care discharge score, nutritional risk score, fall risk score

## Abstract

Background: Harmful in-hospital falls with subsequent injuries often cause longer stays and subsequently higher costs. Early identification of fall risk may help in establishing preventive strategies. Objective: To assess the predictive ability of different clinical scores including the Post-acute care discharge (PACD) score and nutritional risk screening score (NRS), and to develop a new fall risk score (FallRS). Methods: A retrospective cohort study of medical in-patients of a Swiss tertiary care hospital from January 2016 to March 2022. We tested the ability of the PACD score, NRS and FallRS to predict a fall by using the area under curve (AUC). Adult patients with a length of stay of ≥ 2 days were eligible. Results: We included 19,270 admissions (43% females; median age, 71) of which 528 admissions (2.74%) had at least one fall during the hospital stay. The AUC varied between 0.61 (95% confidence interval (CI), 0.55–0.66) for the NRS and 0.69 (95% CI, 0.64–0.75) for the PACD score. The combined FallRS score had a slightly better AUC of 0.70 (95% CI, 0.65–0.75) but was more laborious to compute than the two other scores. At a cutoff of 13 points, the FallRS had a specificity of 77% and a sensitivity of 49% in predicting falls. Conclusions: We found that the scores focusing on different aspects of clinical care predicted the risk of falls with fair accuracy. A reliable score with which to predict falls could help in establishing preventive strategies for reducing in-hospital falls. Whether or not the scores presented have better predictive ability than more specific fall scores do will need to be validated in a prospective study.

## 1. Background

Falls are a common problem in hospital settings with rates of 7 to 17 falls per 1000 patient days in geriatric rehabilitation wards [1] and up to 10 falls per 1000 patient days in internal medicine wards [2]. Even though only 1 to 3% of falls cause a severe injury such as a fracture, minor injuries may also cause prolonged hospitalization and delayed discharge when further assessment of fall consequences is needed [3]. Falls also lead to a loss of confidence and distress among mostly older patients [4]. One way to prevent in-hospital falls would be to identify in-patients at high risk of falling early on and start using preventive strategies [1].

Numerous risk assessment tools for fall risk prediction including measuring physical mobility based on walking speed in a timed up and go test (TUG) [5], mental state, and independence in living at home do exist [1]. All these tools have only moderate discriminative ability; for instance, the TUG has a reported area under curve (AUC) of 0.57 (95% CI, 0.54–0.59) [6] and the Morse fall scale (MFS) has an AUC of 0.60 (95% CI, 0.51–0.68) [7], and these are usually not part of routine care, putting an additional strain on personnel [1]. Therefore, the use of routine tools to predict in-hospital falls would be highly efficient. In fact, several routine scores are potential candidates for predicting falls. For instance, malnourished in-patients are 8 times more likely to fall compared to patients without malnutrition [8,9]. As the nutritional risk score (NRS) is used for billing purposes in Swiss hospitals, it is gathered at every admission, and its information can further be leveraged to predict falls [10]. Another score that is routinely recorded in many Swiss hospitals is the post-acute care discharge (PACD) score, which estimates the probability of the need for assistance in activities of daily living after discharge [11]. It has been shown that a fall in the last 12 months markedly increased the risk of a consecutive fall in many multifactorial fall risk assessments where functional ability in daily activities plays a major role as a predictor [12,13]. Hence, we postulated that the PACD score as a measure of the functional ability to carry out ADL may correlate well with fall risk as well.

So far, there are no studies looking at the association of the NRS, PACD score and in-hospital falls. In this study, we analyzed the predictive value of the NRS and PACD score in predicting in-hospital falls and developed a combined score including NRSs and PACD scores.

## 2. Methods

This is a hospital based retrospective cohort study conducted at the Medical University Clinic of Kantonsspital Aarau, a public, tertiary care, 600-bed teaching hospital in Switzerland. We used electronic health record (EHR) data from January 2016 to March 2022 from medical emergency admissions who did not reject general consent. We excluded all patients aged 18 years or less and admissions with a length of in-hospital stay of less than 2 days. This report follows the TRIPOD criteria for prediction model validation [14].

### 2.1. Outcome

The primary outcome was an in-hospital fall. The secondary outcome was injury severity after an in-hospital fall divided into 4 categories: no, minor, moderate, or severe injury. Minor injury, including bruises, does not need medical treatment. Moderate injury requires little medical attention such as dressing or stitches. Severe injury includes broken bones or head wounds [15].

### 2.2. Instruments

The NRS is composed of 3 clinical categories: impaired nutrition status (0 to 3 points), severity of disease (0 to 3 points) and an age of >70 (0 or 1 point) [10], giving a maximum score of 7 points for the highest risk of malnutrition (see Appendix A). In clinical use, a score of 3 points warrants the optimization of nutritional intake [10].

The PACD score is composed of age, the number of active medical problems and the amount of self-care and daily living activities at home before admission [16,17] (see Appendix A). A score of 8 points or more indicates a high risk of discharge to a post-acute care facility [18].

In a sub-analysis (see Appendix A), we analyzed the delirium observation screening scale (DOS) which was not included in our combined score due to there being a large portion of missing data (see Appendix A). The DOS consists of 13 items of behavioral and cognitive observations [19] and has high reliability in the early detection of delirium [20,21].

### 2.3. Statistics, Sample Size and Missing Data

Data are presented as medians or as counts and frequencies where appropriate. The dataset was split according to the admission time. Data from 2016 to 2019 were used to develop the models (60%) while data from 2020 to March 2022 were used for validation (40%). We did not perform a formal sample size calculation. We performed logistic regression analysis to predict the risk of in-hospital fall.

We fitted an unadjusted model between the outcome and the main predictors, the NRS and PACD score. We then also developed a new fall risk score (FallRS), by linearly combining both scores into an overall score. In a sub-analysis, an unadjusted model between the DOS and fall probability and a further model adjusted for age, gender, the Elixhauser comorbidity index [22], and the Frailty score [23] (see Appendix A).

To calculate the area under curve (AUC), the predicted probabilities were entered into a non-parametric model to assess their discriminatory power. The severity of fall was modeled in an ordinal logistic regression and adjusted for age, gender, the Elixhauser comorbidity index and death in the hospital. To facilitate interpretation, we calculated the marginal probabilities at distinct cutoffs. We conducted a complete case analysis, and calculations were performed using Stata 15 (StataCorp, College Station, TX, USA). All tests were two-sided at an alpha level of 5%.

## 3. Results

We included 19,270 patients (43% females; median age, 71) from 22,366 admissions of which 528 (2.74%) patients had at least one fall during the hospital stay (Figure 1). The baseline characteristics of the patients included in our analysis are presented in Table 1. Admissions with falls were older and had a higher LOS, Elixhauser comorbidity index and Frailty score. Additionally, those who fell were less likely to live at home independently before admission and had a higher rate of in-hospital death.

### 3.1. Calibration

Calibration was assessed by tabulating the observed risk against each point of a score and its predicted risk (Appendix A). Calibration was modest on average.

### 3.2. Discrimination

AUCs are presented in Table 2. The discrimination of validation values revealed that they were modest and ranged from 0.61 (95% confidence interval (CI), 0.55–0.66) for the NRS to 0.69 (95% CI, 0.64–0.75) for the PACD score and 0.70 (0.65–0.75) for the FallRS. The AUC of the DOS was 0.63 (95% CI, 0.60–0.67) (see Appendix A). The adjusted models showed only slightly better discrimination compared to the crude models (see Appendix A).

For clinical practice, an easy-to-remember cutoff with a low number of false positives is key, otherwise scarce resources are spread over too many cases. Thus, we defined a desired specificity of 75%. At this specificity, a PACD score of 11 could identify admissions with fall with a sensitivity of 43%, a FallRS of 13 had a sensitivity of 49% and a NRS of 4 had a sensitivity of 29% (see Table 3).

We analyzed the severity of fall using an ordinal logistic regression and calculated the marginal probability of a harmful fall. There was no correlation between increasing score values and the severity of fall sequalae in any of the scores (see Table 4).

## 4. Discussion

Our large retrospective cohort study of medical in-patients at a tertiary care center in Switzerland found that the combination of two already established scores for malnutrition and post-acute care demand show high potential in predicting in-hospital falls. Additionally, we combined the two scores into a fall risk score (FallRS) which is easy to use and may help in identifying in-patients with an increased risk of falling, initiating early fall prevention strategies [1].

In 2017, a systematic review reported that the detection of fall risk combined with specific interventions can lead to a reduction in inpatient falls by 20–30% [4]. Not only are patients without a fall more self-confident and active after discharge [4], but reducing the fall rate is cost-effective as in-hospital falls may prolong the length of stay [3]. Our data confirmed the association of increased fall risk with older age and increased comorbidities (measured using the Elixhauser comorbidity score and frailty score) as described by multiple other studies [3,4,15]. The NRS and PACD score, as well as our FallRS tended to be higher in those experiencing a fall, which may be an expression of muscle weakness and of a lower capability of selfcare as prescribed as risk factors for falls in other studies [3,4]. Patients with a fall were less likely to live at home independently before admission which corresponds to the older age of this group. Roughly 95% of patients experiencing a fall sustained no or only minimal physical harm such as bruises and did not need medical treatment. This may be explained with the assessment of falls, as we also monitored minor accidents such as slipping out of bed. Overall, only 5% of patients were moderately to severely injured and needed medical care such as stitches and dressings because of lacerations (moderate injury), or even a cast or operation because of fractures or head wounds (severe injury) [15]. We saw a higher frequency of in-hospital deaths in those who fell. However, as only 2% of all falls in our cohort suffered a severe injury, we interpreted this as a chance finding that was most probably attributed to the higher comorbidity in this group. Furthermore, our analyses showed that the severity of fall-induced injuries did not correlate with increasing levels of a score. A prediction of the severity of a fall is therefore not possible with the scores examined. To the best of our knowledge, there is no published score predicting fall severity.

A systemic review and meta-analysis conducted by Barry et al. from 2014 found that the limited discriminative performance with an AUC of 0.57 (95% CI 0.54–0.59) for the specific TUG task used was lower than that revealed by our findings [6]. The PACD score and FallRS had a similar AUC (0.69 (95% CI, 0.64–0.75) and 0.70 (95% CI, 0.65–0.75), respectively) while the NRS was lower with an AUC of 0.61 (95% CI 0.55–0.66). Compared to the MFS, the PACD score and FallRS are more accurate predictors of fall risk as the predictive ability of the MFS is reported to have an AUC of 0.60 (95% CI, 0.51–0.68) [7]. In our cohort, the overall discriminative ability of the unadjusted models was fair and ranged from an AUC of 0.61 (95% CI, 0.55–0.66) for the NRS to an AUC of 0.70 (95% CI, 0.65–0.75) for the FallRS. The adjusted models showed only slightly better discrimination, not adding substantial clinical benefit.

As we had a lot of missing data on the DOS, we decided not to include it in the FallRS.

For clinical practice, a cutoff with a high specificity is desirable in reducing false positives. Given these prerequisites, we derived a cutoff for a PACD of 11 points (sensitivity, 43%), a FallRS of 13 points (sensitivity, 49%), and a NRS of 4 points (sensitivity, 29%). As the PACD and FallRS show similar ability to predict in-hospital fall, we recommend using the simple PACD score rather than the combined FallRS for fall prediction assessment and initiating early fall prevention strategies. This might lead to lower costs via the shorter in-hospital stays and fewer diagnostic assessments and treatments needed after a fall [1,3].

We analyzed the correlation between fall severity and values of each score. In our analysis, higher scores were not associated with more harmful falls.

### Strengths and Limitations

Our study has some important limitations. Falls are a rare event (2.74% of admissions in our cohort) making them difficult to predict. We analyzed only data of one hospital, and as such our data are only moderately generalizable. In our cohort, not every score was systematically assessed in every patient. A direct comparison with a gold standard in fall prediction such as the TUG or MFS was not possible because we do not assess TUG and MFS regularly in our patients. The scores used in this study have not been tested extensively for reliability and validity. To the best of our knowledge, there are currently no studies looking at the reliability and validity of the PACD score. There is one study currently being conducted, but results have not been published yet [24]. We only found one study suggesting the good validity of the NRS when comparing it against commonly used measures of malnutrition such as subjective global assessment, body mass index, calf circumference, bioelectrical impedance analysis-derived phase angle, and hand grip strength [25]. However, this analysis did not report any data on reliability. Additionally, we have no data concerning follow-up events after discharge. In our study, we focused on maximizing the rate of true-positive cases to optimize in-hospital patient processes and save the use of hospital resources for potential false-negative cases. As a limitation, we accept a low level of specificity. Having a rare outcome is good for the patient but makes it difficult to research and document improvements. To test the validity of the PACD score as a fall predictor, it needs to be tested in a prospective setting.

Our work also has several strengths. First, we analyzed a large cohort of 19,270 admissions over multiple years and applied validated, widely used scores. In our hospital, we use an objective protocol of falls, where the documented injuries are signed by both nurses and physicians. For this study, the TRIPOD criteria for prediction model validation [14] were used to standardize the formal processes and presentation. Additionally, there are no studies reporting an association between the TUG score or the MFS and injury severity after a fall. We are the first study to look into the association of fall severity with fall prediction scores. The time period during which we collected data is comparable to that of other studies [3]. In our cohort, there were more male admissions than there were in other studies [2,9] and length of stay was shorter [1,2]. Overall, our cohort was similar to that of other European studies on in-hospital fall. Furthermore, we had less harmful falls than other studies did [1,9]. A recent international study for global guidelines on falls prevention and management in older adults by Montero-Odasso et al. from 2022 stresses the importance of a person-centered approach and emphasizes the importance of physical activity [12], both of which are covered by the PACD score. In comparison to other studies, we established a new perspective on use of these scores which has not been provided before. Until now, the PACD score has not been assessed as a fall prediction tool. As such, we provide an innovative approach to reusing data from established scores, reducing the burden on hospital personnel.

## 5. Conclusions

Even though it was not originally designed for this purpose, the PACD score has a good predictive ability which is even better than that of dedicated scores developed to predict fall risk. The easy-to-compute PACD could easily be implemented in any hospital’s daily routine to assess the fall risk of in-patients at admission. It is very quick to acquire and needs less time than more elaborate tools such as the TUG or MFS (e.g., TUG 5 min [6] vs. PACD 1 min (internal reference)) do.

## Figures and Tables

**Figure 1 geriatrics-08-00060-f001:**
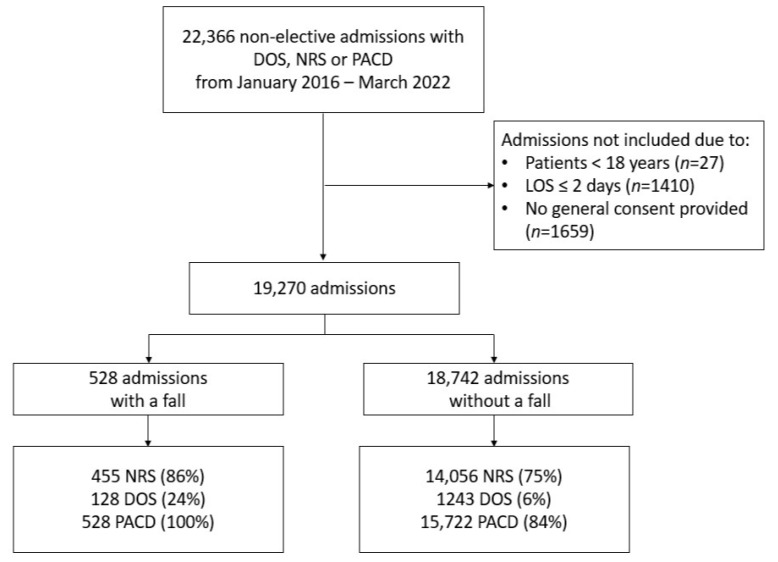
Study flow chart. Abbr.: NRS (nutritional risk screening score), DOS (delirium observation screening scale), PACD (post-acute care discharge), and LOS (length of stay).

**Table 1 geriatrics-08-00060-t001:** Baseline characteristics of the study cohort.

Scores		Total	No Fall	Fall	*p*-Value
		*N* = 19,270	*N* = 18,742	*N* = 528	
NRS		2 (1–3)	2 (1–3)	3 (2–4)	<0.001
PACD score		5 (3–10)	5 (3–10)	10 (5–14)	<0.001
DOS		0 (0–1)	0 (0–1)	1 (0–3)	<0.001
FallRS		8 (5–12)	8 (4–12)	12 (8–17)	<0.001
Sociodemographics					
Age		71 (58–80)	71 (58–80)	76 (68–83)	<0.001
Female		8337 (43.3%)	8120 (43.3%)	217 (41.1%)	0.31
Length of stay		6 (4–10)	6 (4–10)	12 (8–19)	<0.001
Number of ICD-10 diagnoses		8 (5–12)	8 (5–12)	13 (10–18)	<0.001
Elixhauser comorbidity index		2 (1–4)	2 (1–4)	4 (3–5)	<0.001
Frailty score		2.7 (0.8–5.5)	2.6 (0.8–5.4)	5.8 (3.45–8.75)	<0.001
Frailty Score	Low < 5	13,776 (71.5%)	13,561 (72.4%)	215 (40.7%)	<0.001
	Intermediate 5–15	5226 (27.1%)	4936 (26.3%)	290 (54.9%)	
	High > 15	268 (1.4%)	245 (1.3%)	23 (4.4%)	
Health insurance					
Supplementary private		3448 (17.9%)	3344 (17.8%)	104 (19.7%)	0.31
Mandatory basic only		15,821 (82.1%)	15,397 (82.2%)	424 (80.3%)	
Discipline					
Internal Medicine		6677 (34.6%)	6473 (34.5%)	204 (38.6%)	
Cardiology		4059 (21.1%)	3996 (21.3%)	63 (11.9%)	
Oncology		2646 (13.7%)	2536 (13.5%)	110 (20.8%)	
Neurology		1863 (9.7%)	1812 (9.7%)	51 (9.7%)	
Gastroenterology		1617 (8.4%)	1578 (8.4%)	39 (7.4%)	<0.001
Pneumology		1148 (6.0%)	1121 (6.0%)	27 (5.1%)	
Nephrology		788 (4.1%)	760 (4.1%)	28 (5.3%)	
Rheumatology		472 (2.4%)	466 (2.5%)	6 (1.1%)	
Place before admission					
Home		14,864 (77.1%)	14,502 (77.4%)	362 (68.6%)	<0.001
Other hospital		1563 (8.1%)	1521 (8.1%)	42 (8.0%)	
Nursing home		1152 (6.0%)	1104 (5.9%)	48 (9.1%)	
Other institutions		971 (5.0%)	927 (4.9%)	44 (8.3%)	
Home with nursing assistance		716 (3.7%)	684 (3.6%)	32 (6.1%)	
Place after discharge					
Home		12,456 (64.6%)	12,297 (65.6%)	159 (30.1%)	
Rehabilitation		4485 (23.3%)	4213 (22.5%)	272 (51.5%)	
Death in hospital		1079 (5.6%)	1017 (5.4%)	62 (11.7%)	<0.001
Home with nursing assistance		799 (4.1%)	770 (4.1%)	29 (5.5%)	
Missing and others		451 (2.4%)	451 (2.4%)	6 (1.1%)	
Consequences of fall					
No injury		366 (1.9%)	N/A	366 (69.3%)	
Minimal injury		133 (0.7%)	N/A	133 (25.2%)	
Moderate injury		19 (0.1%)	N/A	19 (3.6%)	
Severe injury		10 (0.1%)	N/A	10 (1.9%)	

Comparison between “no fall” and “fall” columns only; continuous variables were compared via ANOVA, and categorical variables were compared via Pearson’s chi-squared test. Data are presented as medians (IQR) for continuous measures, and as *n* (%) for categorical measures. Abbr: NRS (nutritional risk screening score; range, 0–7), DOS (delirium observation screening scale; range, 0–13), PACD (post-acute care discharge; range, 3–infinite). Most falls caused no injury (69.3%), and a quarter had minimal injury not needing medical treatment (25.2%). Few suffered moderate injury (3.6%), requiring stitches or a dressing. Only 10 falls had severe consequences (1.9%) such as broken bones or head injuries. N/A, not applicable.

**Table 2 geriatrics-08-00060-t002:** Predictive abilities for in-hospital fall of the three scores. Derivation years 2016–2019 and validation years 2020–2022.

Scores	Derivation 2016–2019 AUC (95% CI)	*N*	Validation 2020–2022 AUC (95% CI)	*N*	*p*-Value
NRS	0.65 (0.61–0.68)	7005	0.61 (0.55–0.66)	2156	<0.001
PACD	0.69 (0.66–0.72)	7005	0.69 (0.64–0.75)	2156	0.81
FallRS	0.71 (0.68–0.74)	7005	0.70 (0.65–0.75)	2156	0.32

Abbr.: AUC (area under curve), CI (confidence interval), NRS (nutritional risk screening score), PACD (post-acute care discharge), FallRS (fall risk score). Legend: *p*-value tests the null hypothesis of AUC derivation equal to AUC validation.

**Table 3 geriatrics-08-00060-t003:** Overall sensitivity, specificity, and discriminatory value of scores. Upper and lower score values with an approximate specificity of 75%. Prevalence of falls in our cohort was 2.74%.

Score, Points	Sensitivity (95% CI, %)	Specificity (95% CI, %)	AUC (95% CI, %)
NRS 3	54.2 (48.4–60.0)	67.4 (66.4–68.4)	0.61 (0.58–0.64)
NRS 4	29.6 (24.5–35.2)	83.2 (82.4–84.0)	0.56 (0.54–0.59)
PACD 10	50.5 (44.7–56.3)	75.1 (74.5–76.0)	0.63 (0.60–0.66)
PACD 11	43.4 (37.7–49.3)	79.4 (78.5–80.2)	0.61 (0.59–0.64)
FallRS 12	54.9 (49.0–60.6)	72.3 (71.4–73.3)	0.64 (0.61–0.67)
FallRS 13	49.2 (43.3–55.0)	76.7 (75.8–77.6)	0.63 (0.60–0.66)

Abbr.: AUC (area under curve), CI (confidence interval), NRS (nutritional risk screening score), DOS (delirium observation screening scale), PACD (post-acute care discharge), FallRS (fall risk score).

**Table 4 geriatrics-08-00060-t004:** Overall severity of fall sequela compared to score values using ordinal logistic regression. Each score is shown at its minimum, median and maximum value.

	Probability of No Injury (%) (95% CI)	Probability of Minimal Injury (%) (95% CI)	Probability of Moderate Injury (%) (95% CI)	Probability of Severe Injury (%) (95% CI)
NRS 0	44.3 (22.1–66.5)	22.9 (14.1–31.6)	3.2 (0.8–5.6)	1.4 (0–2.9)
NRS 2	70.2 (64.4–76.1)	24.6 (19.4–29.9)	3.5 (1.4–5.7)	1.6 (0.2–3.0)
NRS 7	64.2 (46.5–81.8)	29.1 (15.9–42.4)	4.6 (0.5–8.7)	2.1 (0–4.5)
PACD 0	48.7 (29.3–68.0)	42.0 (27.5–56.5)	6.4 (0–13.0)	2.9 (0–7.3)
PACD 5	55.6 (43.3–67.8)	37.3 (26.6–48.0)	5.0 (0.5–9.4)	2.2 (0–5.2)
PACD 25	79.1 (62.0–96.3)	18.6 (3.8–33.4)	1.6 (0–3.9)	0.6 (0–1.8)
FallRS 0	44.3 (22.1–66.5)	44.8 (29.3–60.3)	7.5 (0–15.5)	3.5 (0–8.9)
FallRS 8	55.6 (44.0–67.3)	37.3 (26.9–47.7)	4.9 (0.6–9.3)	2.1 (0–5.1)
FallRS 30	80.6 (63.9–97.3)	17.4 (2.9– 31.9)	1.5 (0–3.5)	0.6 (0–1.6)

Abbr.: CI (confidence interval), NRS (nutritional risk screening score), DOS (delirium observation screening scale), PACD (post-acute care discharge), FallRS (fall risk score). Legend—minimal injury: no medical treatment needed. Moderate injury: little medical treatment, e.g., dressing, stitches. Severe injury: head injuries or broken bones.

## Data Availability

The datasets generated and/or analyzed during the current study are not publicly available due to Swiss law (data sharing is restricted to the country) but are available from the corresponding author on reasonable request if compliant with Swiss law.

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
