# Peer review of "Prediction of In-Hospital Falls Using NRS, PACD Score and FallRS: A Retrospective Cohort Study"

_geriatrics, 2023, doi:10.3390/geriatrics8030060_

Round 1

Reviewer 1 Report

This is a study of the use of different clinical scores to predict in-hospital falls. Strengths of the study include the large number of patients assessed and the sound statistical analyses. Weaknesses include the retrospective design, the poorly developed rationale for selection of clinical tests as predictors, and the limited clinical significance of the results.

The authors provide adequate background information about the problem of in-hospital falls and the benefits of early identification of inpatients who are at high risk of falling. However, the selection of specific clinical tests as predictors is not clearly justified. Of the many measures  associated with risk of falls, the authors chose two, the NRS and the PACD. The logical argument for nutritional risk as a predictor of falls seems especially weak. In addition, it appears that the range of scores on the NRS was very small, making it difficult to show any association between that measure and falls.

The results of the study indicate that patients who fell were older, had a longer length of stay, had more comorbidities, and were less likely to live at home independently prior to admission. None of these findings is new or surprising. The sensitivity and specificity of the measures included in the study were low for all the various cut-off values examined. According to Power et al[1], “For a test to be useful, sensitivity+specificity should be at least 1.5 (halfway between 1, which is useless, and 2, which is perfect).” The combinations of sensitivity and specificity values for the measures used in this study are all far below 1.5. The finding that fall severity is not correlated with scores on the NCRS or PACD may be of some interest to clinicians.

Adding together the NRS and PACD scores to form a new Fall Risk Score (FallRS) did not contribute meaningfully to prediction. Consequently, the authors should remove statements about using the FallRS for fall prediction assessment (lines 199 – 202). It does not make sense to administer two measures (the NRS and the PACD) when the same predictive value can be achieved with only one measure (the PACD).

 The manuscript has a few instances of unclear wording and/or typographical errors that should be corrected.  

Page

Line

Specific Comments

3

97-98

This is not a complete sentence. Did you fit an unadjusted model between fall outcome and DOS, and another model adjusted for age, gender, etc.?

3

99

The numbering of the tables does not appear to be correct in the supplementary materials provided.

3

117

Please add the minimum and maximum possible score for each of the measures in Table 1. In particular, what is the maximum score for the DOS (this is not reported in the text either).

4

128

Calibration results appear to be shown in supplementary tables 1-4, not just table 1.  

4

132

You appear to reporting AUC validation values in this paragraph, but the AUC of 0.58 for the DOS is for derivation (should be 0.63 for validation according to supplementary table 5).  

5

143

The text reports that a FallRS of 8 had a sensitivity of 49%, but table 3 indicates that this was for a FallRS of 13.

6

198

Same issue regarding FallRS of 8 (probably an error in table 3).

[1] Power M, Fell G, Wright M. Principles for high-quality, high-value testing. Evid Based Med. 2013 Feb;18(1):5-10. doi: 10.1136/eb-2012-100645. Epub 2012 Jun 27. PMID: 22740357; PMCID: PMC3585491.

Author Response

This is a study of the use of different clinical scores to predict in-hospital falls. Strengths of the study include the large number of patients assessed and the sound statistical analyses. Weaknesses include the retrospective design, the poorly developed rationale for selection of clinical tests as predictors, and the limited clinical significance of the results.

Reply: Thank you very much for taking your time in reviewing our work. We hope we were able to fully address your concerns.

The authors provide adequate background information about the problem of in-hospital falls and the benefits of early identification of inpatients who are at high risk of falling. However, the selection of specific clinical tests as predictors is not clearly justified. Of the many measures associated with risk of falls, the authors chose two, the NRS and the PACD. The logical argument for nutritional risk as a predictor of falls seems especially weak. In addition, it appears that the range of scores on the NRS was very small, making it difficult to show any association between that measure and falls.

Reply: Thank you for your input. The NRS is a score that is used in Swiss hospitals and some other countries in Europe for billing purposes. As the information is already gathered and we wondered whether or not this readily available data can be further leveraged to save working time. We agree that the NRS is a simple risk stratification with a small range of the score. However, we believe that in this case simplicity is an advantage, as it allows to easily stratify patients into groups. We have adapted the section in the introduction according to your advice as follows:

“…, the use of routine tools to predict in-hospital falls would be highly efficient. In fact, several routine scores are potential candidates to predict falls. For instance, malnourished in-patients are 8 times more likely to fall compared to patients without malnutrition (Eglseer, Hoedl et al. 2020, Ishida, Maeda et al. 2020). As the Nutritional Risk Score (NRS) is used for billing purposes in Swiss hospitals, it is gathered at every admission, and its information can further be leveraged to predict falls (Kondrup 2003). Another score that is routinely recorded in many Swiss hospitals is the Post-Acute Care Discharge (PACD) score, which estimates the probability of need for assistance in activities of daily living after discharge (Louis Simonet, Kossovsky et al. 2008). It has been shown that a fall in the last 12 months markedly increased the risk for a consecutive fall in many multifactorial falls risk assessments where functional ability in daily activities plays a major role as predictor (Montero-Odasso, 2022, Barthel- Index (Mehony and Barthel, 1965). Hence, we postulated that PACD as a measure of functional ability of ADL may correlate well with fall risk as well.”

The results of the study indicate that patients who fell were older, had a longer length of stay, had more comorbidities, and were less likely to live at home independently prior to admission. None of these findings is new or surprising. The sensitivity and specificity of the measures included in the study were low for all the various cut-off values examined. According to Power et al [1], “For a test to be useful, sensitivity and specificity should be at least 1.5 (halfway between 1, which is useless, and 2, which is perfect).” The combinations of sensitivity and specificity values for the measures used in this study are all far below 1.5.

Reply: Thank you for your observation. We agree on Power’s et al. assessment in general, especially when it comes to further testing which might be invasive and costly. On the other hand, our report describes the situation where data is already at hand and re-utilized and now additional testing is carried out. Additionally, we have a situation with a very low pre-test probability which severely hampers predictive performance (analogous to Power et al.: “Prevalence critically affects predictive values.”). In light of these facts, we wanted to optimize the identification of true positive cases and lower false positive as much as possible why we deliberately focused on a high degree of specificity in order to optimize the patient process, but acknowledge that this approach has certain limitations. Thus, we have added this paragraph to the limitations section of the discussion:

“In our study, we focused on maximizing the rate of true positive cases to optimize in-hospital patient processes and save use of hospital resources on potential false negative cases. As a limitation, we accept a low level of specificity. Having a rare outcome is good for a patient but makes it difficult to research and document improvements. To test the validity of the PACD score as a fall predictor, it needs to be tested in a prospective setting.”

The finding that fall severity is not correlated with scores on the NRS or PACD may be of some interest to clinicians.

Reply: Thank you for your comment, we have added the following to the discussion:

“Furthermore, our analyses showed that the severity of fall induced injuries did not correlate with increasing levels of a score. A prediction of the severity of a fall is therefore not possible with the scores examined. To the best of our knowledge, there is no score being able to predict fall severity.”

Adding together the NRS and PACD scores to form a new Fall Risk Score (FallRS) did not contribute meaningfully to prediction. Consequently, the authors should remove statements about using the FallRS for fall prediction assessment (lines 199 – 202). It does not make sense to administer two measures (the NRS and the PACD) when the same predictive value can be achieved with only one measure (the PACD).

Reply: Thank you for this observation to which we fully agree. Hence, we have adapted the Discussion accordingly:

“As PACD and FallRS show similar ability to predict in-hospital falls, we recommend to use the simple PACD score rather than the combined FallRS for fall prediction assessment and initiating early fall prevention strategies. This might lead to lower costs by shorter in-hospital stays and fewer diagnostic assessments and treatments after a fall (David Oliver 2010, Hars, Audet et al. 2018).”

The manuscript has a few instances of unclear wording and/or typographical errors that should be corrected.

Reply: Thank you for your feedback. The sections mentioned have been adapted, see table.

Page

Line

Specific Comments

Corrections

3

97-98

This is not a complete sentence. Did you fit an unadjusted model between fall outcome and DOS, and another model adjusted for age, gender, etc.?

“In a sub-analysis, we fitted an unadjusted model between DOS and fall probability and a further model adjusted for age, gender, Elixhauser comorbidity index (Elixhauser A 1998), and Frailty score (Rockwood, Song et al. 2005) (see supplementary table 4).”

3

99

The numbering of the tables does not appear to be correct in the supplementary materials provided.

Thank you for catching this error.

3

117

Please add the minimum and maximum possible score for each of the measures in Table 1. In particular, what is the maximum score for the DOS (this is not reported in the text either).

Thank you for this input, we added the range of the scores to the description of table 1 as follows:
“Abbr: NRS (nutritional risk screening, range 0 - 7), DOS (delirium observation screening scale, range 0 - 13), PACD (post-acute care discharge, 3 - infinite), As another reviewer also asked for the composition of the scores, you will find a more detailed declaration of all scores in the supplementary material.”

4

128

Calibration results appear to be shown in supplementary tables 1-4, not just table 1.  

Thank you for catching this error.

4

132

You appear to reporting AUC validation values in this paragraph, but the AUC of 0.58 for the DOS is for derivation (should be 0.63 for validation according to supplementary table 5).  

Thank you for your observation. We have corrected this value according to the table 5 as follows:

“The AUC of DOS was 0.63 (95% CI 0.60 – 0.67) (see supplementary table 4).”

5

143

The text reports that a FallRS of 8 had a sensitivity of 49%, but table 3 indicates that this was for a FallRS of 13.

Thank you for catching this error.

6

198

Same issue regarding FallRS of 8 (probably an error in table 3).

Thank you for your observation. We checked table 3 and the calculations are correct. Probably the error was taken from the text above and not noticed. We have corrected this error in the text.

David Oliver, F. H., Terry P Haines (2010). "Preventing falls and fall-related injuries in hospitals." Clinics in Geriatric Medicine.2010 Nov;26(4):645-92.

Eglseer, D., et al. (2020). "Malnutrition risk and hospital-acquired falls in older adults: A cross-sectional, multicenter study." Geriatr Gerontol Int 20(4): 348-353.

Elixhauser A, S. C., Harris DR, Coffey RM (1998). " Comorbidity measures for use with administrative data." Med Care. 1998;36(1):8–27.

Hars, M., et al. (2018). "Functional Performances on Admission Predict In-Hospital Falls, Injurious Falls, and Fractures in Older Patients: A Prospective Study." J Bone Miner Res 33(5): 852-859.

Ishida, Y., et al. (2020). "Malnutrition at Admission Predicts In-Hospital Falls in Hospitalized Older Adults." Nutrients 12(2).

Kondrup, J. (2003). "Nutritional risk screening (NRS 2002): a new method based on an analysis of controlled clinical trials." Clinical Nutrition 22(3): 321-336.

Louis Simonet, M., et al. (2008). "A predictive score to identify hospitalized patients' risk of discharge to a post-acute care facility." BMC Health Serv Res 8: 154.

Rockwood, K., et al. (2005). "A global clinical measure of fitness and frailty in elderly people." CMAJ 173(5): 489-495.

Reviewer 2 Report

Thank you for the opportunity to review the manuscript titled: Prediction of in-hospital falls by NRS, PACD Score and FallRS 2: A retrospective cohort study

The methods used to develop the new falls risk tool are sounds and results are presented well.

However, authors need to validate the FallRS 2 further to conclude that it is able to predict falls and is easier to use (due to combining two other tools). As mentioned in the study, a minority of patients had in-hospital fall and hence this maybe not be the best sample to test its predictive capacity.

Author Response

The methods used to develop the new falls risk tool are sounds and results are presented well.

Reply: Thank you very much for taking your time in reviewing our work.

However, authors need to validate the FallRS 2 further to conclude that it is able to predict falls and is easier to use (due to combining two other tools). As mentioned in the study, a minority of patients had in-hospital fall and hence this maybe not be the best sample to test its predictive capacity.

Reply: Thank you for your observation. We have adapted accordingly:

“In our study, we focused on the highest possible rate of true positive cases to optimize in-hospital patient processes with our existing data set. As a limitation, we accepted a low level of specificity by using already existing data. The resulting low prevalence is good for the patient but leads to difficulties in research and in documenting improvements. To test the PACD score as a fall predictor, the score would need to be tested in a prospective setting.”

Reviewer 3 Report

Thank you for the opportunity to review this research article. The feedback below is intended to strengthen the manuscript.

-Given the importance of the reframing aging initiative, use the term "Older adults" rather than "elderly" in the manuscript

-World guidelines for falls prevention were published in 2022 (Montero-Odasso et al., 2022) and include suggestions for in hospital falls risk assessment. It would be helpful to include this reference and describe what your study adds given the suggestions in these guidelines. 

-Remove the comment "would be interesting" in line 53.

-The TUG is not a measure of strength, this should be corrected in the introduction. 

-Include a description of the psychometrics for the NRS and PACD.

-A researcher with expertise in measurement development is needed to review the method used to create the FallRS 

Author Response

Thank you for the opportunity to review this research article. The feedback below is intended to strengthen the manuscript.

Reply: Thank you very much for taking your time and your valuable feedback.

Given the importance of the reframing aging initiative, use the term "Older adults" rather than "elderly" in the manuscript

Reply: As suggested, we have rephrased the manuscript accordingly.

World guidelines for falls prevention were published in 2022 (Montero-Odasso et al., 2022) and include suggestions for in hospital falls risk assessment. It would be helpful to include this reference and describe what your study adds given the suggestions in these guidelines. 

Reply: Thank you for pointing out this helpful reference. We have included it in the introduction section:

“Another score that is routinely recorded in many Swiss hospitals is the Post-Acute Care Discharge (PACD) score, which estimates the probability of need for assistance in activities of daily living after discharge (Louis Simonet, Kossovsky et al. 2008). It has been shown that a fall in the last 12 months markedly increased the risk for a consecutive fall in many multifactorial falls risk assessments where functional ability in daily activities plays a major role as predictor (Montero-Odasso, 2022, Barthel- Index (Mehony and Barthel, 1965). Hence, we postulated that PACD as a measure of functional ability of ADL may correlate well with fall risk as well.”

As well as in the limitations and strengths section:

“A recent international study for global guidelines on falls prevention and management in older adults by Montero-Odasso et al from 2022 stresses the importance of a person-centred approach and emphasises the importance of physical activity (Montero-Odasso, van der Velde et al. 2022), both of which are covered by the PACD. In comparison to other studies, we established a new perspective on use of these scores which has not been done before. Until now, PACD has not been assessed as a fall prediction tool. As such, we provide an innovative approach to reuse data from established scores reducing the burden for hospital personnel.”

Remove the comment "would be interesting" in line 53.

Reply: Thank you for your comment, we have adjusted the sentence accordingly.

The TUG is not a measure of strength, this should be corrected in the introduction. 

Reply: Thank you for your input. We have adapted as follows:

“Numerous risk assessment tools for fall risk prediction including measuring physical mobility based on walking speed in a Timed Up and Go test (TUG) (Podsiadlo D 1991), mental state, and independence in living at home do exist (Hars, Audet et al. 2018).”

Include a description of the psychometrics for the NRS and PACD.

Reply: Thank you for your valuable advice. We have added the exact measurement parameters of all scores in the supplementary materials (see supplementary tables 1-3). The psychometrics of the NRS and PACD are as follows:

Composition of the NRS as presented in the original study (Kondrup 2003)

Nutritional status

Score

Normal

-

0

Mild

weight loss > 5% in 3 months

OR

food intake below 50-75% of requirement in preceding week

1

Moderate

weight loss > 5% in 2 months

OR

food intake below 25-50% of requirement in preceding week

OR

BMI 18.5-20.5

2

Severe

weight loss > 5% in 1 month

OR

food intake below 0-25% of requirement in preceding week

OR

BMI < 18.5

3

Severity of disease

Normal

-

0

Mild

e.g. hip fracture, diabetes

1

Moderate

e.g. pneumonia

2

Severe

e.g. head injury, intensive care patients

3

Age

>70 years

1

Abbr: NRS (Nutritional Risk Score), BMI (Body Mass Index)

Legend: Add the Score points for a maximum score of 7

Composition of PACD as presented in the original study (Conca, Gabele et al. 2018)

Number of active medical problems on admission

Infinite

Person in the same household who can provide help

Yes = 0 points, No = 4 points

Max. 4 points

Number of limitations in ADL (1 point per limitation)

Needed help for:

-        Personal hygiene

-        Dressing/undressing

-        Toileting

-        Bathing/showering

-        Eating/drinking

-        Moving

-        Transfer (bed/chair)

-        Car or public transportation

-        Shopping

-        Cooking

-        Homework

-        Medication management

Max. 12 points

Age (years)

<60 = 0 points

≥60 = 1 point

≥70 = 2 points

≥80 = 3 points

≥90 = 4 points

≥100 = 5 points

Max. 5 points

Abbr: PACD (Post Acute Care Discharge Score), ADL (Activities of Daily Living)

Legend: Add all points for the score of your patient. Because there is no maximum number of medical problems, the score potentially ranges to infinity.

A researcher with expertise in measurement development is needed to review the method used to create the FallRS.

Reply: Thank you for your hint. Professor Schuetz, Professor Müller and Dr. Struja have wide experience in researching clinical scoring systems and also designed numerous scores. A non-exhaustive list of references is attached below. We hope this fully addresses your concerns.

https://pubmed.ncbi.nlm.nih.gov/34799814/

https://pubmed.ncbi.nlm.nih.gov/34003435/

https://pubmed.ncbi.nlm.nih.gov/31965531/

https://pubmed.ncbi.nlm.nih.gov/31023276/

https://pubmed.ncbi.nlm.nih.gov/30791054/

https://pubmed.ncbi.nlm.nih.gov/30647051/

https://pubmed.ncbi.nlm.nih.gov/28005916/

https://pubmed.ncbi.nlm.nih.gov/28100628/

Conca, A., et al. (2018). "Prediction of post-acute care demand in medical and neurological inpatients: diagnostic assessment of the post-acute discharge score - a prospective cohort study." BMC Health Serv Res 18(1): 111.

Hars, M., et al. (2018). "Functional Performances on Admission Predict In-Hospital Falls, Injurious Falls, and Fractures in Older Patients: A Prospective Study." J Bone Miner Res 33(5): 852-859.

Kondrup, J. (2003). "Nutritional risk screening (NRS 2002): a new method based on an analysis of controlled clinical trials." Clinical Nutrition 22(3): 321-336.

Louis Simonet, M., et al. (2008). "A predictive score to identify hospitalized patients' risk of discharge to a post-acute care facility." BMC Health Serv Res 8: 154.

Montero-Odasso, M., et al. (2022). "World guidelines for falls prevention and management for older adults: a global initiative." Age Ageing 51(9).

Podsiadlo D, R. S. (1991). "The timed "Up & Go": a test of basic functional mobility for frail elderly persons." Journal of American Geriatric Society.

Round 2

Reviewer 3 Report

The authors have addressed the feedback provided with the exception of providing psychometrics (reliability and validity statistics) on the PACD and NRS.

Author Response

Reviewer 3 – Second rebuttal letter

Question: The authors have addressed the feedback provided with the exception of providing psychometrics (reliability and validity statistics) on the PACD and NRS.

Reply: Thank you very much for your advice and sorry our misunderstanding. Initially, we did not understand what was exactly meant by psychometrics. We have now searched PubMed extensively for psychometrics (reliability and validity) of the NRS and the PACD.

Regarding the PACD, our collaborators Conca et al. proposed a validation study in 7 Swiss hospitals in 2021. However, the results are not yet available (Conca, Koch et al. 2021). As far as we know, there are no further studies on this topic at the moment. This could be due to the fact that the score is still relatively novel and is not yet as established internationally as other scores.

Regarding the NRS, in a study conducted by Speranza in 2022 the NRS appears to be a valid tool for assessing malnutrition at hospital admission compared to other malnutrition scores (Subjective Global Assessment (SGA), the Global Leadership Initiative on Malnutrition (GLIM) criteria and Handgrip Strength (HGS)) (Speranza, Santarpia et al. 2022).

In general, studies using the NRS have very diverse cohorts leading to a large variation in the average score values. For example, an NRS <3 was found in 33% (Zacharias and Ferreira 2017), 65% (Chen, Li et al. 2022), and 93% (Speranza, Santarpia et al. 2022) of medical patients in studies investigating medical emergency admission, stroke, and liver transplants, respectively.

Unfortunately, we could not find studies testing the reliability of the NRS.

Hence, we have added the following sentences to the limitation section:

“The scores used in this study have not been tested extensively for reliability and validity. To the best of our knowledge, there are currently no studies looking at the reliability and validity of the PACD. There is one study currently being conducted, but results have not been published yet (Conca, Koch et al. 2021). We only found one study suggesting good validity of the NRS when comparing it against commonly used measures of malnutrition such as subjective global assessment, body mass index, calf circumference, bioelectrical impedance analysis-derived phase angle, and hand grip strength (Speranza, Santarpia et al. 2022). However, this analysis did not report any data on reliability.”

Chen, X., et al. (2022). "Nutritional risk screening 2002 scale and subsequent risk of stroke-associated infection in ischemic stroke: The REMISE study." Front Nutr 9: 895803.

Conca, A., et al. (2021). "Self-Care Index and Post-Acute Care Discharge Score to Predict Discharge Destination of Adult Medical Inpatients: Protocol for a Multicenter Validation Study." JMIR Res Protoc 10(1): e21447.

Speranza, E., et al. (2022). "Nutritional Screening and Anthropometry in Patients Admitted From the Emergency Department." Front Nutr 9: 816167.

Zacharias, T. and N. Ferreira (2017). "Nutritional risk screening 2002 and ASA score predict mortality after elective liver resection for malignancy." Arch Med Sci 13(2): 361-369.